# Drone-based vertical profiling of particulate matter size distribution and carbonaceous aerosols: urban vs. rural environment

Kajal Julaha[1,2,*], Vladimír Ždímal[2], Saliou Mbengue[3], David Brus[4], Naděžda Zíková[2,5,*]

[1]Department of Atmospheric Physics, Faculty of Mathematics and Physics, Charles University, Prague, 18000, Czech Republic.
[2]Institute of Chemical Process Fundamentals of the Czech Academy of Sciences, Prague 16500, Czech Republic.
[3]Global Change Research Institute of the Czech Academy of Sciences, Brno 60300, Czech Republic.
[4]Atmospheric Composition Research, Finnish Meteorological Institute, Helsinki 00560, Finland.
[5]Institute for Environmental Studies, Faculty of Sciences, Charles University, Prague, 12801, Czech Republic

*Correspondence to: zikova@icpf.cas.cz (N. Zíková), julaha@icpf.cas.cz (K. Julaha)

**Keywords:** Equivalent black carbon, vertical distributions, Drone, micro-aethalometer AE51, optical particle counter, Aethalometer AE33, optical particle sizer, humidity control.

**Abstract.** The study presents drone-based measurements to investigate the seasonal vertical variability of equivalent black carbon (eBC) mass and particle number concentrations (PNC) at a rural and urban site in the Czech Republic. Vertical profiles of eBC were measured using a micro-aethalometer, while PNC was measured using an optical particle counter. Drone-based eBC measurements closely matched reference aethalometers placed at both ground level and at 230m of a tower when using a humidity control mechanism. Without dryer, eBC mass concentration was overestimated by 276% in summer and 285% in winter, but uncertainties were reduced to under 10% with drying. These findings highlight the importance of humidity control for accurate aerosol measurements, especially for eBC. The study also revealed a decrease in eBC and PNC with height at the rural site during both summer and winter, with seasonal differences in the altitude where this decrease began. Elevated eBC concentrations in winter were due to increased atmospheric stability and combustion-related fine particles. At the urban site, concentrations in summer were uniform with height (4 to 100 m AGL) but gradually decreased with height during winter. Furthermore, the study investigated changes in the vertical distribution of eBC and PNC during a high pollution event at the urban site, influenced by long-range transport. Our findings confirm the effectiveness of drones in capturing vertical variations of air pollutants, offering results on the dynamics between local emissions, atmospheric stability, and long-range transport and suggesting the necessity of measuring vertical concentration profiles to support air quality management strategies.

## 1. Introduction

Black Carbon (BC) aerosols, one of the substantial contributors to climate change and adverse health effects, are primarily emitted into our atmosphere through incomplete combustion of fossil fuels and biomass (Bond et al., 2013; Ramanathan and Carmichael, 2008). BC absorbs efficiently solar radiation and contributes to atmospheric warming (Moteki, 2023; Myhre et al., 2013). Aged BC can act as cloud condensation nuclei (CCN) and affect climate through its indirect effects by altering cloud properties and their formation processes (Wang et al., 2018c). The radiative properties of BC depend on its vertical profiles (Samset et al., 2013). For example, BC in the free troposphere can enhance its radiative forcing by trapping energy emitted from the lower cloud layers (Schwarz et al., 2006). The vertical distribution of BC also impacts the evolution of the planetary boundary layer (PBL). BC

in the upper PBL exhibits light absorption efficiency, heating the surrounding atmosphere and enhancing
atmospheric stability, leading to extreme haze pollution events (Ding et al., 2016).
Modeling-based studies on BC vertical distribution are limited (Chen et al., 2022). Uncertainties in these
models mainly arise from assumptions about the vertical distribution of BC aerosols, highlighting the need to
measure the vertical distribution of BC on a regional scale, from areas influenced by direct emissions from the
ground to those characterized by long-range transport (Ramana et al., 2010). These measurements can also help
validate satellite observations and improve the representation of BC vertical profiles in climate models, leading
to a more accurate assessment of BC radiative forcing (Li et al., 2013; Samset et al., 2013).
The BC vertical distribution can be measured by various platforms, such as meteorological balloons,
towers, aircraft, and unmanned aerial vehicles (UAVs). Meteorological tethered balloons provide highly resolved
data and detailed information close to the ground, capable of measuring aerosol concentrations up to the free
atmosphere (Babu et al., 2011; Ferrero et al., 2019; Renard et al., 2020; Cappelletti et al., 2022). Meteorological
towers offer a unique opportunity for continuous long-term monitoring of aerosols at different heights (Chi et al.,
2013; Xie et al., 2019; Sun et al., 2020; Liang et al., 2022). Compared to towers, aircraft and UAVs can access
higher altitudes, with some aircraft capable of carrying heavier payloads, allowing them to transport more
sophisticated instruments for detailed aerosol measurements. These platforms offer greater spatial coverage and
flexibility, making them suitable for comprehensive atmospheric studies (Brady et al., 2016; Corrigan et al., 2007;
Villa et al., 2016; Wu et al., 2021; Schulz et al., 2019). Drones have recently gained popularity among all the other
methods because of their cost effectiveness, flexibility, and mobility due to their lightweight design (Barbieri et
al., 2019; Boer et al., 2020). Several studies have used drones to study vertical measurements of BC and particle
number concentrations (PNC). For example, Liu et al. (2020) conducted vertical measurements of fine particulate
matter (PM) and BC using a DJI Matrice 600 drone equipped with a battery-operated light-scattering laser
photometer and a micro-aethalometer. Their study revealed different vertical patterns for PM2.5 and BC,
suggesting different sources for each. Similarly, Zhu et al. (2019) used a hexacopter with a customized scanning
mobility particle sizer, an optical particle counter, and a meteorology sensor to study the vertical variability of
particle number size distribution (PNSD) near the ground to up to 300 m. The study showed that PNC with size
>0.3 μm decreased with height during the evening. Brus et al. (2021) investigated the vertical profile of PNCs and
gases in the San Luis Valley, Colorado, and highlighted their interaction with meteorological conditions and
boundary layer processes. Studies on the vertical distribution of BC aerosols in Central Europe are very limited.
In Poland, Chilinski et al. (2016) examined the vertical distribution of BC in a valley for three days using UAV.
In Germany, Samad et al. (2020) investigated the vertical profiles of PM, BC, and ultrafine particles in Stuttgart
using a tethered balloon, and Harm-Altstädter et al. (2024) used a fixed-wing drone for vertical measurement of
aerosol concentration, including eBC, near a civil airport.
The studies about the vertical distributions of BC aerosols in the Czech Republic are limited to a tall
tower in a rural area (Mbengue et al., 2023), and no measurements in urban areas have been done. To date, no
drone-based measurement of BC has been conducted in the Czech Republic. This study combines mobile (drone-
based) and fixed (tall tower and building) observational platforms to measure the vertical distribution of BC
aerosols and PNC at two different sites representing an urban and a rural location to isolate the respective roles of
local emissions, meteorology, and long-range transport in shaping vertical aerosol distributions. It further
estimates the measurement uncertainties and dependence of the results on the humidity. In this study, we address
this gap by developing and testing a lightweight, drone-mountable silica-gel dryer that enables humidity-
controlled eBC measurements.

## 2. Materials and Methodology

### 2.1. Measurement Sites

#### 2.1.1. Rural background site

The National Atmospheric Observatory Košetice (NAOK, 49°35′N, 15°05′E; 534 m a.s.l.) in the Bohemian Moravian Highlands in the Czech Republic (Figure 1) represents a central European background site. Located approximately 75 kilometers southeast of Prague, the observatory is situated in a rural area. The observatory is equipped with instruments to measure gaseous pollutants, atmospheric aerosols, and meteorological parameters. It includes a 250 m tall atmospheric tower which provides a unique opportunity to study atmospheric parameters at different elevations (Dvorská et al., 2015). NAOK is part of the Aerosol, Clouds, and Trace Gases Research Infrastructure Network (ACTRIS ERIC) and several other research projects and monitoring programs (Mbengue et al., 2023).

NAOK is influenced by regional and long-range transported air masses, mainly associated with the western and southeastern directions (Mbengue et al., 2021; Vodička et al., 2015). A primary highway in the Czech Republic (D1: 36,000 cars/day, CSD, 2020) is situated approximately 6 km to the north and northeast of the observatory (Mbengue et al., 2023).

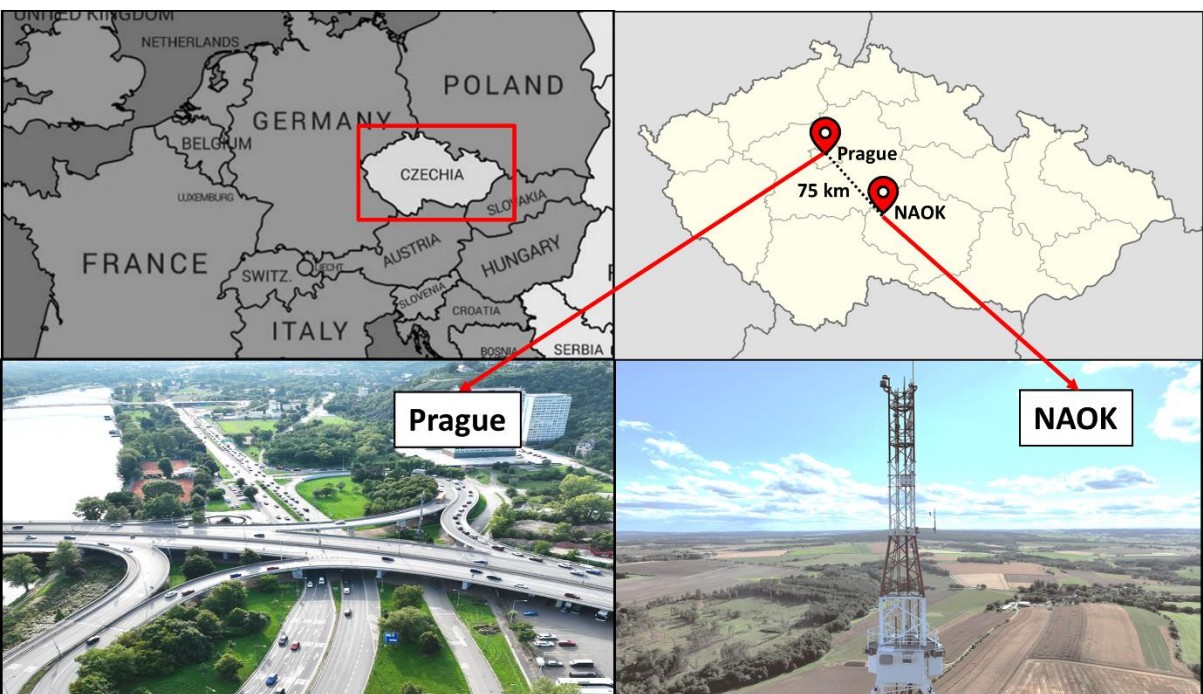

**Figure 1. Geographical location of rural background (NAOK) and urban (Prague) sites in the Czech Republic. Source: mapchart.net, Wikipedia.**

### 2.1.2. Urban site

The Faculty of Mathematics and Physics (50°6.89′N, 14°26.95′E; 185 m a.s.l.) at Charles University in Prague represents an urban site situated 75 kilometers away from NAOK (Figure 1). The faculty has multiple campus locations throughout Prague, with its Troja campus near the Vltava River serving as the site for this study. The main building of the Troja campus is an 11-story building, almost 50 m high. The campus is located in a river valley surrounded by hills with an elevation of 50 m AGL. The Department of Atmospheric Physics (DAP) is also on this campus. The DAP monitors particulate matter ($PM_1$, $PM_{2.5}$, and $PM_{10}$), gases ($NO_2$, $O_3$, and CO), and meteorology (temperature, relative humidity (RH), pressure, wind speed, and rainfall), with measurements taken at ground level (2m), 10 m, and 50 m (Ramatheerthan et al., 2024).

The site is located near the Blanka tunnel exit and is impacted by fresh traffic emissions. The Blanka tunnel, more than 6 km long, is the longest road tunnel in the Czech Republic. The average traffic density of this tunnel is 80,000 to 90,000 cars/day (Metrostav, 2024). It was constructed to minimize the environmental impacts of traffic. However, its opening significantly increased the traffic at some locations, leading to substantial changes in the urban geochemistry of Prague (Mizera et al., 2022).

### 2.2. Instrumentation

### 2.2.1. eBC measurements

The micro-Aethalometer AE51 (AethLabs San Francisco, CA) (Figure 2a) provides real-time equivalent BC (eBC) concentration using an 880 nm light source (Petzold et al., 2013). AE51 operates on a principle similar to other aethalometers, such as AE31 (Aerosol Magee Scientific, Berkeley, CA). The AE51 measures the light attenuation through a filter (T60 Teflon-coated glass fiber) loaded with particles and converts the attenuation into an eBC mass concentration using a predefined mass attenuation coefficient (12.5 $m^2/g$) (Alas et al., 2020). The time resolution of 10 seconds and flow rate of 150 ml/min were used in this study. The filter was replaced when attenuation, a dimensionless measure of optical absorbance, reached 80 to minimize the filter loading effect. This threshold has been recommended to reduce measurement bias due to increasing filter loading (Good et al., 2017; Lee, 2019; Miyakawa et al., 2020). Recent studies (Alas et al., 2020; Masey et al., 2020) have shown that uncorrected AE51 readings closely match reference instruments in low-concentration environments. Therefore, no correction method was used for the eBC values in the present study.

To reduce short-term noise, AE51 data were averaged over 1-minute intervals. If the resulting mean eBC was negative, the measurement point was excluded from further analysis. The fraction of excluded data was below 2% for all periods except the NAOK summer, where it reached a maximum of 10% at altitudes above 100 m.

### 2.2.2. Air stream Dryer

A 20 cm-long homemade silica gel dryer (Figure 2d) was used in front of the AE51 (Figure 2f) to control the humidity for accurate eBC mass concentration measurements. The dryer consists of 2 coaxial cylinders of 1.62 cm and 0.65 cm diameters, with silica gel in the space between them. The silica gel effectively removes moisture from the aerosols as axial airflow passes through the dryer. The silica gel used in the dryer was spherical bead-type (Carl Roth, P077.1, "Perfform"), which is mechanically robust and non-dusting, minimizing any risk of particle shedding under vibration. Additionally, the dryer was sealed with stainless-steel mesh (inner cylinder) at

both ends to prevent the possible release of silica fragments during operation. The inner cylinder (diameter 6.6 mm and length 13.5 cm), of stainless steel woven mesh screen with a 0.25 mm x 0.25 mm square hole aperture (80 opening per inch, 0.05 mm wire, ~65 % open area), was chosen for its smooth surface and minimal particle loss, while the outer parts were fabricated with PLA (Polylactic Acid) using a 3D printer (MK4S, Prusa Research), with a total weight of 50 g. Particle loss was evaluated using the Particle loss calculator (von der Weiden et al., 2009) and found to be ≤1% for PM2.5-sized particles at the AE51's flow rate of 150 mL/min (Table S2). The performance of the dryer was tested in laboratory conditions by passing air with 100% RH through the setup at the AE51's flow rate of 150 mL/min. The dryer effectively reduced the RH to below 40%, and maintained that level for up to 3 days, ensuring reliable drying under operational flow (Figure S1). The silica-gel beads were replaced every morning before measurements began. The flow and leakage tests were also carried out to describe the dryer's performance at 150 mL/min. The flow rate was monitored before and after the dryer using a mass flow meter. For the leak test, the dryer inlet was connected to a HEPA filter, and the outlet was connected to a Condensation Particle Counter (CPC) to monitor any particle breakthrough. Particle concentrations measured by the CPC were found to be negligible, confirming the air-tight integrity of the dryer assembly.

### 2.2.3. Particle number concentration measurements

The air quality measurements backpack (Yugen Oy, Finland) for a consumer-grade drone with an Optical particle Counter (OPC-N3, Alphasense) (Figure 2b) was used to measure PNC in the polystyrene latex (PSL) equivalent size range from 0.35 to 40 µm. The OPC detects the light scattered by particles in the sample air stream illuminated by a laser beam (~658 nm) and translates the signal into particle count and size (Hagan and Kroll, 2020). The OPC-N3 reports an internal airflow estimate based on a low-power internal fan performance, not corrected for external wind. The OPC's inlet was horizontally mounted and exposed to wind during drone flights so that it faced oncoming airflow. While this minimized directional variability, strong horizontal winds could still affect the internal airflow stability of the OPC-N3 (Table S2). To mitigate this, all measurements were averaged over 1-minute intervals, which helps reduce short-term fluctuations. Due to OPC's horizontal inlet design and a low power built-in ventilator, equipping a dryer would result in an excessively high pressure drop (manufacturer's maximum allowable pressure drop ≤ 40 Pa), making the measurement highly unreliable (Bezantakos et al., 2020) and thus the OPC-N3 was operated without a dryer. Since OPC-N3 sampled air without drying, the measured particle sizes and number concentrations may therefore be affected by hygroscopic growth under high relative humidity conditions. However, the instrument's internal T is slightly elevated due to electronics heat emission, reducing the humidity of the sampled air. Analysis of the internal RH logs revealed that no data exceeded 80% RH, and most measurements were taken under relatively dry conditions (RH < 40% in 60–90% of cases, depending on the season and height). Therefore, hygroscopic growth effects were expected to be minor. Similar limitations and evaluation strategies (flagged RH > 80 %) have been documented in previous UAV-based OPC studies (Brus et al., 2025; Chacón-Mateos et al., 2022; Nurowska et al., 2023; Nurowska and Markowicz, 2023)."

The backpack with OPC uses a Raspberry Pi zero microcomputer as a data logger and was mounted on the top of the drone (Figure 2g). The backpack also contains two meteorological sensors BME 280 (Bosch Sensortec GmbH) and SHT85 (Senserion AG) positioned on opposite side of the backpack (see their comparison in the next section) and a redundant to drone own GPS module for the recording of drone position (Brus et al., 2025). The backpack housing was 3D-printed using white polyethylene terephthalate glycol (PETG) filament,

which provides structural support and helps reflect solar radiation to minimize thermal influence on the sensors. The dual-sensor configuration also reduces bias caused by asymmetric solar heating, which can lead to small temperature differences (up to a few degrees) under clear-sky conditions, while remaining negligible under overcast skies. Temperature and RH readings from both sensors were compared against tower-based temperature and RH data while flying on the drone at different heights to evaluate the feasibility and reliability of using the drone-based setup for vertical profiling of temperature and RH validate sensor accuracy and data reliability (Figure S2-S5).

The total particle number concentration ($N$), in particles per cubic meter (#/m$^3$), was calculated from the raw OPC data as:

$$N = \frac{C}{F \cdot t},$$ (1)

where $C$ is the total particle count, $F$ is the flow rate in cm$^3$/s, and $t$ is the sampling time in seconds. The OPC operates at a total flow rate of 5.5 l/min and a sample flow rate of 0.28 l/min. The measurement interval of 1 second was used to account for the high temporal variability of particles' concentrations.

**2.2.4. Temperature and RH**

For OPC, sensor SHT85 was used to measure ambient temperature (T) and RH, while with AE51, an Arduino (MKR Zero) datalogger (HYT939p, Innovative Sensor Technology IST AG) (Figure 2c) was used for T and RH measurements. Although HYT939p has a slower nominal response time (≈2–3 s), potential lag effects were negligible because the drone hovered for approximately 5 min at each altitude and data were averaged over 1-min intervals. The HYT939P showed close agreement with tower-based T/RH measurements (see Figures S6 and S7 and Table S3), confirming its suitability for UAV-based profiling. The Arduino MKR zero microcontroller processes sensor data using a 32-bit SAMD21 processor and stores it on an SD card. It is programmed via the Arduino IDE to read inputs, perform tasks, and save data. The Arduino datalogger with HYT939p sensor was developed after the first summer campaigns at both sites and, therefore, was used only during winter campaigns. For the summer campaign, meteorological variables from the tower at the same height as the drone hover at NAOK were used, and in Prague, meteorological data from the ground and top of the building, i.e., 50 m, were used.

UAV-based T and RH measurements showed strong agreement with tower observations (R² = 0.85–0.99 for T and 0.50–0.96 for RH; RMSE = 0.3–2.4 °C and 5–8 %, respectively, Figure S2-S7). Among the sensors, the HYT939p exhibited the most stable performance and the smallest bias, while the SHT85 and BME showed largest scatter (Figure S8 and Table S3). On average, UAV readings were ~0.3 °C warmer and ~7 % drier than the tower reference. The moderate reduction in correlation above 150 m, primarily due to fewer data points and a response lag at 230 m, reflects the known limitations of compact airborne sensors (Brus et al., 2025). These uncertainties, however, remain within acceptable limits for UAV-based meteorological measurements, as demonstrated in previous intercomparison studies (Barbieri et al., 2019).

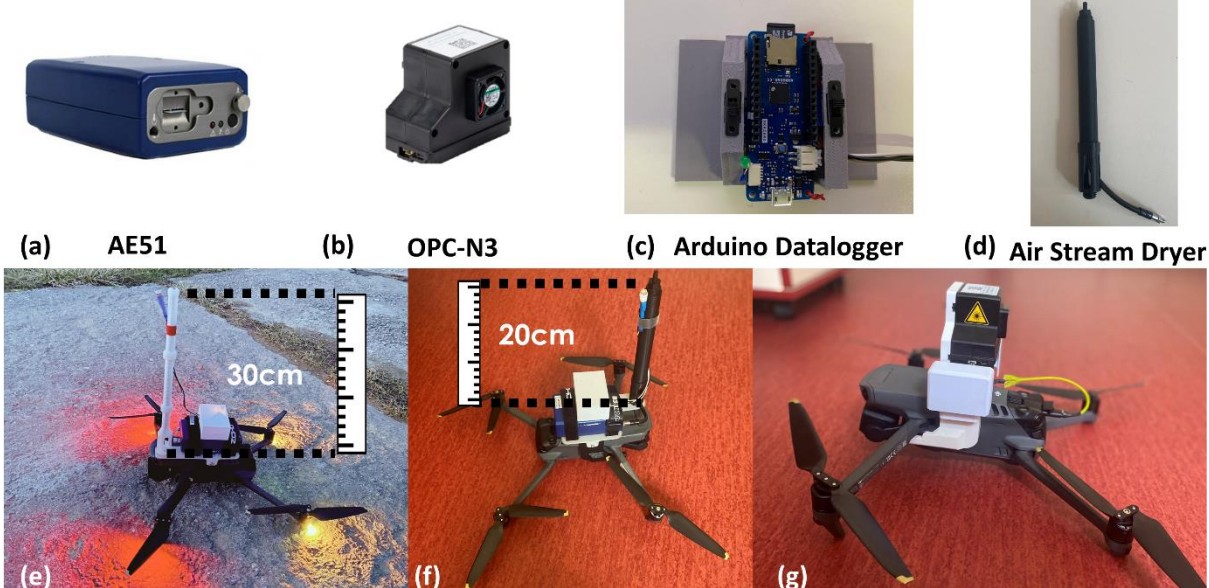

210

**Figure 2. Measurement setup: (a) micro-Aethalometer AE51, (b) optical particle counter (OPC) N3, (c) Arduino datalogger, (d) Air Stream Dryer, (e) micro-Aethalometer AE51 with a temperature and RH datalogger without a dryer, (f) micro-Aethalometer AE51 with a temperature and RH datalogger with a dryer, and (g) consumer drone backpack with an optical particle counter with a custom mount on the drone.**

### 2.2.5. Drone-based sampling

The instruments were installed on the Mavic 3 Classic drone (DJI Techology Co., Ltd.) (https://www.dji.com/cz/mavic-3-classic/specs). The instrument's combined weight was too high to be carried by the drone; thus, the instruments were set up separately and measured each alternating hour under similar meteorological conditions.

With the micro-Aethalometer AE51, two different types of inlets were used: a 30 cm high inlet without a dryer and a 20 cm high inlet with a diffusion-based silica gel dryer, while no inlet was used with a drone backpack (Figure 2). In the dryer-equipped setup, in contrast to the non-dryer configuration using a single bend (Figure 2e), an additional 90° bend in the inlet tubing was necessary to accommodate the dryer housing (Figure 2f). Based on an equivalent pipe length (EPL) of 0.15 m per 90° bend (radius < 5 cm; Wang et al., 2002), the EPL increased from 42 cm (without dryer) to 49 cm (with dryer). The AE51's sampling height and mounting position remained unchanged, and this small difference in EPL is negligible for submicron aerosols at 0.15 L min⁻¹ (von der Weiden et al., 2009).

Particle losses within the AE51 inlet system were estimated using the Particle Loss Calculator (von der Weiden et al., 2009) for all measurement configurations and wind speeds (Table S2). For a 30 cm high inlet without dryer, total transmission efficiency for particles ≤ 2.5 μm corresponded to losses ≤ 9 %, while for the 20 cm dryer inlet, losses were ≤ 1 %. Whole-inlet losses were evaluated for both AE51 configurations (with and without dryer) to assess the effect of wind speed on sampling accuracy. For the AE51 without dryer, concentration changes for $PM_{2.5}$ remained minimal (< 10 % overestimation) up to 4 m s⁻¹ but increased at higher wind speeds, reaching about 22 % overestimation for $PM_1$ at 6 m s⁻¹. When the dryer was attached, the sampling efficiency also

decreased with wind speed, with overestimation increasing from ~5 % at 2 m s$^{-1}$ for PM$_1$ to ~50 % at 6 m s$^{-1}$ for
PM$_{2.5}$.
Only sampling losses were calculated for OPC and OPS, as no inlet extension was used. For OPS,
sampling loss was minimal (10 % overestimation) for PM$_{2.5}$ fraction up to wind speed of 6 m/s, but PM$_{10}$ showed
an underestimation of 100 % up to 6 m/s. For OPC, sampling effects were less severe (50 % overestimation up to
6 m/s) for PM$_1$, but for PM$_{2.5}$, overestimation ranged from 60 % to 125 % at 4 m/s and 6 m/s, respectively. For
PM$_{10}$ particles, overestimation was as high as 750% at 6 m/s.
During the flights, the drone climbed vertically from the ground to 230 m and 100 m AGL at a constant
speed of 1 ms$^{-1}$ along the tower at NAOK and the Prague building, respectively (the maximum altitude was limited
to 100 m in Prague due to flight height restrictions). The drone hovered at different heights (4 m, 50 m, 100 m,
150 m, and 230 m at NAOK and 4 m, 50 m, and 100 m in Prague) for at least 3-5 minutes and then ascended in
the same vertical direction. To reduce short-term noise, the raw data were averaged into 1-minute intervals,
yielding 3–5 values per altitude per flight. Although continuous measurements were recorded during each ascent
flight, only hovering measurements at different heights were used in this study. This approach reduces the
influence of rotor-induced turbulence and enables more stable sampling conditions. While ascent and hovering
show a 3-6 % error in particle concentration, the descending flights were excluded due to the propellers-induced
airflow increasing apparent particle concentrations by up to 40–60% (Hedworth et al., 2022).
Flights were conducted for at least 4 to 5 days during a week, depending on the weather conditions.
Across the full campaign (approximately 15–20 flights), these 1-minute averages were grouped by altitude, and
the resulting distributions were used for comparisons with reference instruments (e.g., AE33) and visualized at
each height level.
To compare OPC-N3 (0.35–37 μm measurement size range) with the OPS (0.3–10 μm size range), size-
bin harmonization was applied. For each 1-min average, OPC binned number concentrations were linearly
interpolated in $\log_{10}(D_p)$ space from OPC mid-bin diameters onto the OPS bin grid over the overlapping size range
of 0.35–10 μm (Weltje and Roberson, 2012). For all other analyses and plots, no bin harmonization was applied
and data are presented in the full size ranges.
The summer campaign took place at NAOK from July and August 2023, and the winter campaign in
February 2024 (Table 1). Additionally, a test to evaluate the dryer's performance was conducted on August 13,
2024. The dryer-on intercomparison at NAOK was performed on a single day. In Prague, measurements were
performed during two summer campaigns and one winter campaign across 2023 and 2024. The urban
measurements were taken without a dryer in August 2023, and in December 2023, while a dryer was used for eBC
measurements in July 2024 (Table 1). Measurements for each campaign began at 06:00 UTC (08:00 CEST) and
continued until 18:00 UTC (20:00 CEST) during the summer, and from 07:00 UTC (08:00 CET) to 16:00 UTC
(17:00 CET) during the winter, due to shorter daylight hours. The number of flights for eBC and PNC at each
height is summarized in Table 1.
**Table 1. Overview of campaign schedule and total number of flights for eBC and PNC measurements. Campaigns with**
**a dryer are indicated with an asterisk.**

| | Season | Campaign Dates | Number of flights (eBC) | Number of flights (PNC) | Measurement height (m AGL) |
|---|---|---|---|---|---|
| NAOK | Summer | July 31 to August 4, 2023 | 20 | 18 | 4,50,100,150,230 |
| | Summer* | August 13, 2024* | 12* | --- | 4*,230* |
| | Winter | February 12 to 16, 2024 | 15 | 15 | 4,50,100,150,230 |
| Prague | Summer | August 14 to 20, 2023 | 22 | 21 | 4,50,100 |
| | Summer* | July 18 to 23, 2024* | 21* | 21* | 4*,50*,100* |
| | Winter | December 12 to 18, 2023 | 17 | 17 | 4,50,100 |

### 2.2.6. Additional variables

At NAOK, two aethalometers AE33 were available as reference instruments: one at the ground with a $PM_{10}$ sampling inlet (Leckel GmbH) at 4 m AGL and the other installed at the top of the tower, i.e., at 230 m with the same sampling head as on the ground. The data from these aethalometers were compared with the drone-based measurements while the drone hovered at corresponding heights. The AE33 at the ground uses a Nafion dryer (custom-made, TROPOS, Leipzig, Germany) to remove moisture from the sample stream, whereas AE33 at 230 m was connected to a Nafion dryer but was not supplied with dry air during the summer of 2023 and winter campaigns. An Optical Particle Sizer (OPS) (model 3330, TSI Inc., USA), without any dryer to ensure similar measurement conditions, was placed at 4 m for comparison with the measurement from OPC on the drone. In addition, temperature, RH, global radiation, wind speed and direction, and gaseous concentrations were obtained from standard measurements at multiple tower heights (50 m, 125 m, and 240 m) and ground level (4 m) (Dvorská et al., 2015), and ceilometer CL51 (Vaisala, Finland) was used for every hour boundary layer height (BLH) information (Julaha et al., 2025) at NAOK..

In Prague, long-term measurements alongside the building include data on temperature, RH, wind speed, gaseous concentrations, and particulate matter concentrations, monitored at ground level, 10 m, and at the top of a 50m high building (Table 2) (Ramatheerthan et al., 2024). Since ground-based ceilometer measurements for the BLH were not available at the site, boundary layer height predictions were obtained from ERA5, a fifth-generation ECMWF (European Centre for Medium Weather Forecasting) reanalysis model produced by the Copernicus Climate Change Service (C3S). The hourly boundary layer height was obtained for the duration of campaigns (Hersbach et al., 2023). The consistency between ERA5-derived and ceilometer-observed BLH values was previously assessed (Julaha et al., 2025), showing good agreement at NAOK. Consequently, ERA5 data were considered reliable for estimating BLH at the urban site.

Normality was evaluated using the Shapiro–Wilk test ($p < 0.05$). Since the data were non-normally distributed, the Kruskal–Wallis test was used to determine significant differences in eBC (and PNC) concentrations between sampling heights. Sample sizes (N) are shown on boxplots. All groups had $N \geq 10$, satisfying the minimum requirement ($N > 5$) for reliable Kruskal–Wallis testing of small, non-normal datasets (Sheskin, 2003).

The percentage difference (PD) was calculated to evaluate variability across measurements for comparing data across different heights and conditions:

$$PD = \frac{X_{ref} - X_{drone}}{X_{ref}} * 100, \tag{2}$$

where $X_{ref}$ is concentration from reference device and $X_{drone}$ is concentration from device on drone. The same
approach was taken also for calculating the difference between heights.

305        The wind shear between the heights was calculated as the difference in wind speed ($\Delta$WS) divided by

the difference in altitude ($\Delta$z):
$$Wind\ Shear = \frac{\Delta WS}{\Delta z}, \tag{3}$$

given in m/s per 100 m.
**Table 2. Variables and instrumentation used in this study.**

| | Instruments | variables | Measurement heights (m AGL) | |
|---|---|---|---|---|
| | | | NAOK | Prague |
| Drone | AethLabs AE51 | eBC | 4, 50, 100, 150, 230 | 4, 50, 100 |
| | Alphasense OPC N3 | PNC | 4, 50, 100, 150, 230 | 4, 50, 100 |
| | BME and SHT85 | T, RH, P | 4, 50, 100, 150, 230 | 4, 50, 100 |
| | Arduino HYT939p | T, RH | 4, 50, 100, 150, 230 | 4, 50, 100 |
| | Drone (DJI Mavic 3 Classic) | ws | 4, 50, 100, 150, 230 | 4, 50, 100 |
| Fixed | Magee AE33 | eBC | 4, 230 | - |
| | TSI OPS 3330 | PNC | 4 | - |
| | Vaisala Ceilometer CL51 | BLH/MLH | ground | - |
| | Tower measurements | T, RH, P, ws, wd | 10, 50, 125, 240 | - |
| | ENVISENS M-22-017 | Global Radiation | ground | 50 |
| | Envitech ED-19-004, ED-19-005 | PM | - | 10, 50 |
| | Aeroqual AQS1, Envitech M-22-016, M-22-017 | NO2, O3, CO | - | 2, 50 |
| | Davis Vantage Pro2, Meteopress MD1017, MD1016 | T, H, P, ws | - | 10, 50 |
| | ERA5 | BLH | - | - |

**3.      Results and Discussion**
**3.1.      Intercomparison and effect of RH on eBC and PNC measurements**
To assess the reliability of drone-based aerosol observations, eBC and PNC measured while hovering the drone
were compared with the observations from the reference devices from the NAOK tower at 4m and 230m for both
the summer and winter campaigns. Because the reference instruments were available only at the NAOK site, the
validation of drone-based measurements based on 15 days of measurements was performed exclusively there,
providing basis for assessing instrument performance under real ambient conditions.

317        The AE51 on the drone overestimated the median reference eBC mass concentration by approximately

276 % at 4m and by 99 % at 230m during summer, with absolute differences of 0.32 µg/m³ and 0.15 µg/m³,

respectively. The smaller overestimation at the 230 m can be due to similar measurement conditions as both AE33 at 230 m and the AE51 on the drone were operating without any dryer (except Nafion without dry air in front of the AE33, which may have partially influenced the moisture content of the sampled air). The higher difference at 4m during the summer is likely due to high RH affecting the eBC measurements; while a Nafion dryer was installed in front of the AE33 on the ground, the AE51 on the drone without a dryer was strongly influenced. This was further confirmed when the RH dropped below 40 % on 3 August, 2023, and eBC mass concentrations from AE51 on the drone were comparable with the reference devices at both 4 m and 230 m, as indicated by Kruskal-Wallis (KW) test showing no significant difference ($p > 0.05$) (Figure S9).

During the winter campaign, median drone-based measurements using AE51 overestimated eBC mass concentrations by 285 % (0.7 µg/m$^3$) at 4 m and by 150 % (0.4 µg/m$^3$) at 230 m compared to the reference AE33 observations (Figure 3). This can be attributed again to the influence of humidity—at 4m, the AE33 was operated with a dryer, and the temperature gradient between inside the measurement container and the external environment at 230 m likely contributed to some drying effects as the sample travelled from the colder outdoor to the warmer indoor environment.

The PNC from OPC on the drone also showed overestimation compared to the OPS reference observations by 75 % (8 #/cm$^3$) and 129 % (30 #/cm$^3$) during summer and winter, respectively. The comparison was made using the same size bins, with the interpolation applied to align the bins between the two instruments. Both the OPC and OPS measurements were conducted without a dryer for both seasons, thus measuring aerosol PNC at ambient RH. The observed difference can be attributed to different sampling orientations: OPC inlet sampled horizontally against the wind, while OPS had a vertical inlet, causing different influence on sampling in both instruments. For OPS, the sampling showed overestimation within 10 % for PM$_{2.5}$ up to wind speed of 6 m/s. In contrast, for OPC, overestimation jumped to 60 % and 125 % for PM$_{2.5}$ at wind speeds of 4m/s and 6m/s, respectively. As a result, OPC tends to report higher particle concentration than the OPS, which contributes to the discrepancies observed in the PNC values. Furthermore, the absence of drying likely enhanced apparent particle sizes during high-RH periods; however, internal OPC-N3 RH records indicate that all measurements were performed at RH < 80 %. The slightly elevated internal temperature of the OPC reduces in-flow humidity, thereby suppressing hygroscopic particle growth. Consequently, humidity-related artefacts were limited and do not affect the interpretation of relative vertical and seasonal variability.

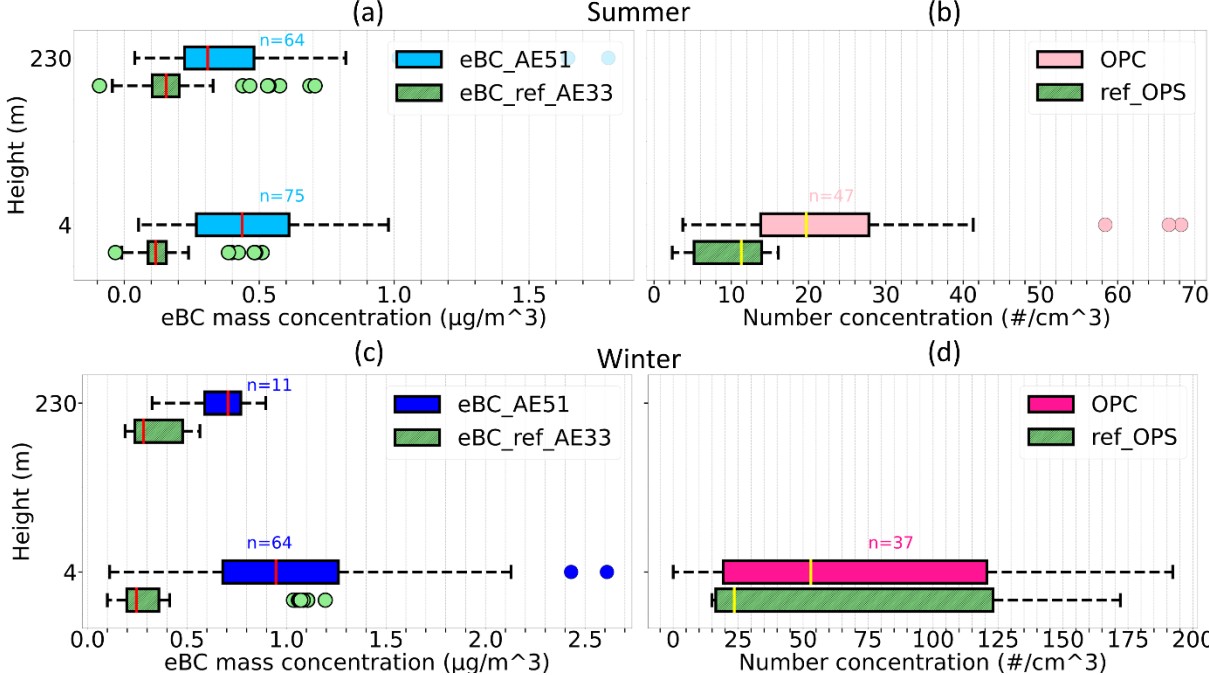

**Figure 3. Boxplots of (a) eBC mass concentration and (b) PNC from drone vs. reference devices in the 0.35 – 10 µm size range from the tower at 4 m and 230 m during the summer 2023 campaign at NAOK; c) and d) the same for winter 2024. Boxes show median and IQR; whiskers extend to 1.5×IQR; points beyond are outliers. n = number of points (1-min means) per altitude.**

To address the effect of RH on eBC concentrations from drone measurements, a homemade silica gel dryer was installed on the drone. A test to evaluate the dryer's performance was conducted on August 13, 2024, a typical summer day with a temperature of 28 °C, RH varying from 50% to 90%, and a wind speed of 2-3 m/s. Additionally, the aethalometer on the top of the tower at NAOK was equipped with a nafion dryer to ensure consistent comparison between the two AE33 at different levels and between AE33 and AE51 under varying RH levels throughout the day. The eBC measurements were done with and without the dryer at the drone and compared to the AE33 eBC concentrations at the tower (both with Nafion dryers).

During this particular summer day, the AE51 on the drone without the dryer overestimated eBC mass concentrations by 29 % (0.09 µg/m$^3$) at 4 m and by 53 % (0.22 µg/m$^3$) at 230 m compared to the reference AE33 (Figure 4a). After installing the silica gel dryer on the drone, the eBC measurements were closely aligned with the reference observations, with the difference reduced to under 10% (0.01 µg/m$^3$ at 4 m and 0.02 µg/m$^3$ at 230 m) at both heights (Figure 4b). This highlights the significant role of the dryer in minimizing the humidity impacts and enhancing the accuracy of eBC mass concentration measurements from the micro-aethalometer AE51. These findings further confirm the reliability of the drone platform and its effectiveness in providing eBC measurements that compare well with long-term tower observations. The AE51, like other single-spot aethalometers, can respond to light-scattering aerosols. This effect was minimized by using a dryer and verified by the close agreement with AE33 data, so any positive bias from non-absorbing particles was considered negligible.

The strong overestimation of eBC by the AE51 under high relative humidity is consistent with hygroscopic growth of scattering aerosols, which increases apparent light attenuation (Cai et al., 2013). Water uptake by soluble particles can amplify both scattering and absorption on the AE51's filter, artificially inflating

the reported eBC. This bias is especially problematic in mobile measurements, where the RH fluctuates rapidly.
By installing a silica-gel dryer to maintain RH below ~ 40 %, the humidity-induced artifact was eliminated, and
AE51 readings aligned well with reference AE33 measurements.

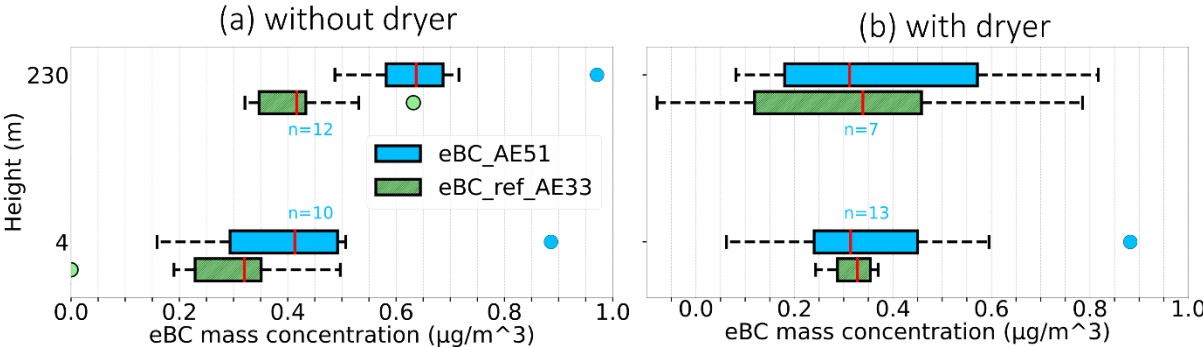


**Figure 4. Boxplots of eBC mass concentration from drone and reference devices on the tower at NAOK during a summer day (August 13, 2024) at 4 m and 230 m (a) without the dryer and (b) with the dryer. Boxes show median and IQR; whiskers extend to 1.5×IQR; points beyond are outliers. n = number of points (1-min means) per altitude.**

## 3.2.     Aerosol vertical profile at the rural site

Vertical profiles of eBC mass concentrations without the silica gel dryer and PNC were measured while hovering
the drone at different heights (4 m, 50 m, 100 m, 150 m, and 230 m) during the summer 2023 and winter 2024 at
NAOK simultaneously with the reference instruments (Figure 5). During summer, eBC mass concentration
remained relatively uniform up to the height of 50 m, followed by a decrease of 32 % (0.13 $\mu g/m^3$) between 50
and 100m. Conversely, PNC dropped by 30 % (6 #/$cm^3$) between 4 m and 50 m. In winter, eBC mass concentration
stayed constant up to 100 m and decreased by 18 % (0.16 $\mu g/m^3$) between 100 m and 150 m. PNCs were constant
from the ground to 50 m but decreased by 39 % (24 #/$cm^3$) between 50 m and 100 m. The significance of the
increase or decrease in eBC mass concentration and PNC was tested by the Kruskal-Wallis (KW) test ($p < 0.05$).
When plotted using native (non-interpolated) bins, the OPC-N3 and OPS show closer agreement (see Figure. 5).

390        The general decrease of both eBC and PNC with height indicates that surface sources dominate aerosol

loading at this rural site and that vertical mixing was insufficient to fully homogenize the boundary layer. Similar
vertical gradients have been observed in background and rural settings, where limited turbulence allows
combustion-derived fine particles to accumulate near the ground, resulting in declining concentrations aloft
(Harm-Altstädter et al., 2024; Samad et al., 2020).

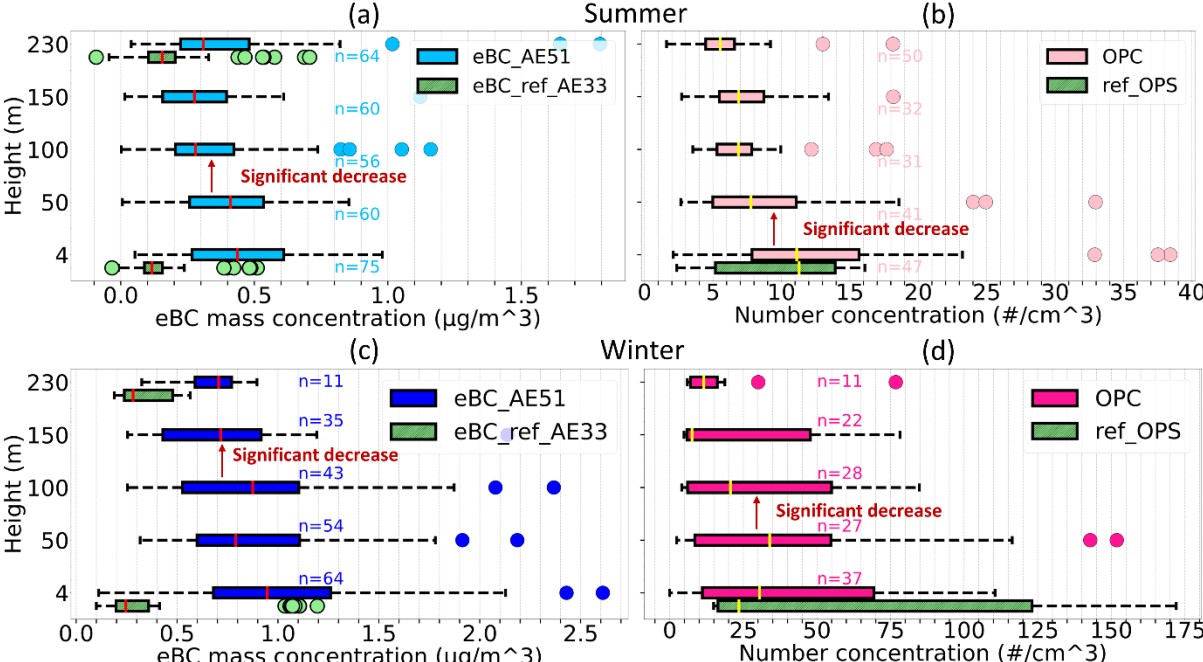

**Figure 5. Boxplots of (a) eBC mass concentration and (b) PNC from the drone (full size range used) while hovering at different altitudes during summer 2023 at NAOK; c) eBC mass concentration and (d) PNC from the drone while hovering at different altitudes during winter 2024 at NAOK. Boxes show median and IQR; whiskers extend to 1.5×IQR; points beyond are outliers. n = number of points (1-min means) per altitude.**

Simultaneously, the vertical gradient of temperature and RH were examined during the winter campaign to explain the vertical changes in eBC and PNC. The eBC, PNC, temperature, and RH comparison revealed no significant temperature variation from the ground to 50m for both eBC mass and PNC (Figure 6). The nearly uniform (isothermal) temperature profile ($\Delta T \approx 0.05$ °C between 4 m and 50 m) indicates very weak vertical temperature gradients and limited turbulent exchange, which can promote accumulation of eBC and PNC near the surface. Such near-isothermal conditions correspond to a shallow, stable mixed boundary layer, typical of winter mornings in mid-latitude regions, when solar heating is too weak to drive convective turbulence and mix the surface air upward (Steeneveld, 2014). Previous studies reported similar near-surface accumulation of pollutants under weak or isothermal temperature (Marucci and Carpentieri, 2019; Wang et al., 2018b).

The temperature started to decrease with height above 50 m, and the PNCs decreased, while eBC mass concentrations remained constant up to 100 m despite the temperature changes. This vertical pattern is similar to the summer measurements, where eBC mass was uniform up to 50 m, and PNC decreased from the ground. This difference between eBC and PNC with altitude likely reflects particle size and lifetime differences: eBC, mostly sub-micron, has lower deposition velocities and longer residence times, while larger or semi-volatile particles dominating PNC are more prone to settling and condensation losses. Deposition velocity increases markedly with particle size (Donateo et al., 2023), making dry deposition a key driver of size-dependent vertical gradients. This indicates that differences in particle size may have brought the observed changes in PNC compared to the unaffected eBC mass concentrations. The consistency across seasons suggests that eBC is well mixed within the lower mixed layer, while PNC is governed more by local production and removal processes such as coagulation and hygroscopic growth.

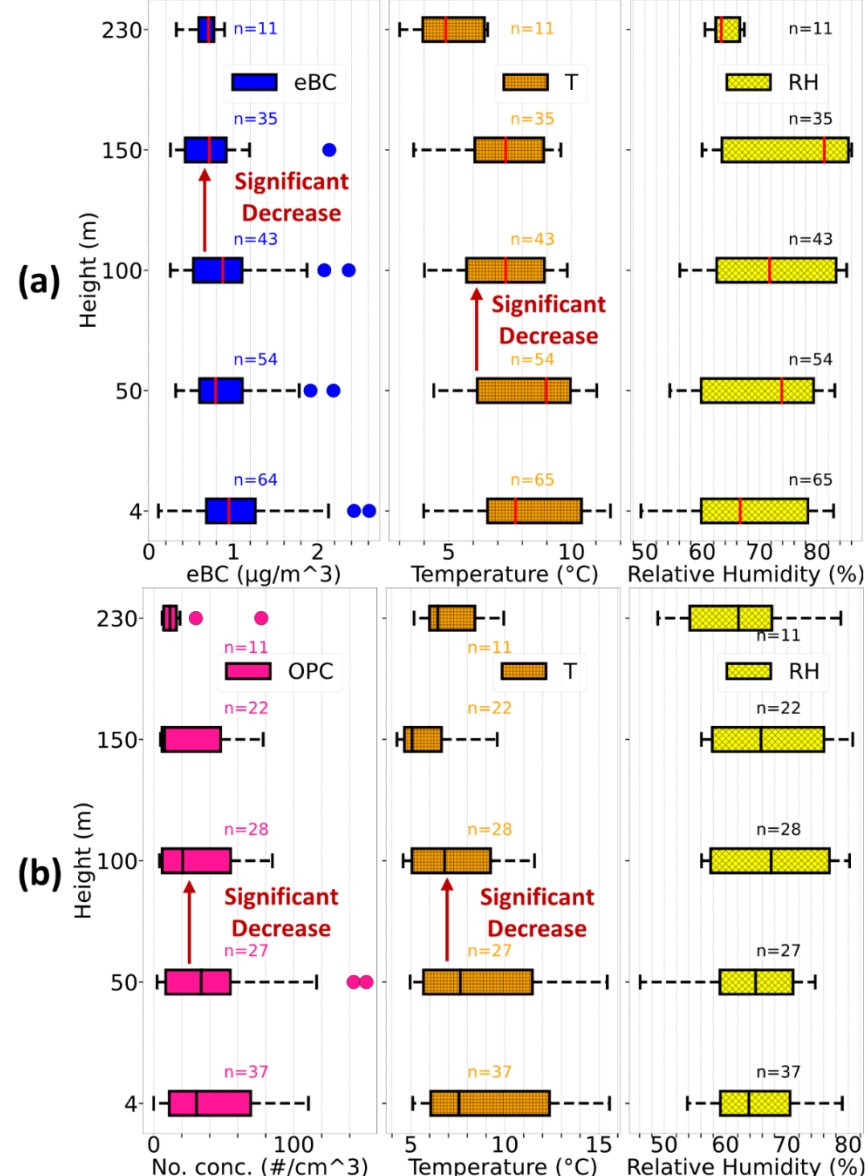

**Figure 6. Boxplots of (a) vertical distribution of eBC mass concentration from AE51 without dryer, temperature, and RH, and (b) vertical distribution of PNC from OPC, temperature, and RH on the drone at rural site NAOK during winter 2024 . Boxes show median and IQR; whiskers extend to 1.5×IQR; points beyond are outliers. n = number of points (1-min means) per altitude.**

Further, the decrease in eBC mass concentration with height was more pronounced in summer (32 %) compared to winter (18 %) at NAOK. On the contrary, PNC decreased with height more during winter (39 %) than in summer (30 %). These contrasting patterns reflect the role of meteorology and emissions: in summer, deeper boundary layers and convective mixing disperse eBC more aloft, whereas in winter, shallow mixing retains it near the surface. The steep winter decline of PNC likely arises because heating emissions emit coarser or semi-volatile particles that are efficiently lost with height, but in summer, new particle formation produces many fine particles that distribute more uniformly (Gao et al., 2012; Kulmala et al., 2004). While vertical mixing influences the vertical distribution of particles, the behavior of eBC vertical distributions reflects the combination of particle size and atmospheric stability rather than primarily depending on vertical mixing alone (Wang et al., 2018). Our results suggest that at least two aerosol populations of different sizes and sources were measured during the year,

thus with different vertical behaviors. This was confirmed by comparison to reference AE33 data from the tower; in winter, biomass/wood burning contributed 48% and 44% of eBC measured at 4 m and 230 m, respectively, leading to a higher absorption Ångström exponent (AAE) of 1.6 and 1.5, respectively, while in summer, fossil fuel combustion (AAE of 1.19 and 1.24, respectively) was the main source of eBC at NAOK, and biomass burning contributing to 18% and 22% of eBC, respectively.

### 3.3.  Aerosol vertical profiles at the urban site

At the urban site, eBC mass concentration and PNC measurements were conducted up to 100 m during summer in two different years – 2023 and 2024. The eBC mass concentrations were measured without the dryer from August 14 to August 20, 2023, and with the dryer from July 18 to Juy 23, 2024. During both summers, eBC mass concentration and PNC were uniform up to the height of 100m (Figure 7). This consistency can be attributed to several factors. The high number of traffic emission sources at the site contributes to high and relatively stable eBC concentrations in the lower atmosphere, similar to the results of (Liu et al., 2023). Also, enhanced thermal convection and the urban heat island effect facilitate effective vertical mixing (Battaglia et al., 2017). Furthermore, wind shear above 2.0 m/s per 100 m between all the heights (4-50m and 50-100m) during both years supports the vertical transport of pollutants. The combination of convective and mechanical turbulence facilitates the rapid vertical redistribution of aerosols, resulting in a uniform eBC and PNC profile despite strong surface emissions. The presence of local sources in the city is further supported by Czech Hydrometeorological Institute (CHMI) ground-based $PM_{10}$ observations from nearby Karlín (traffic site) and Kobylisy (urban background) stations, which show higher concentrations and distinct diurnal peaks consistent with local traffic and resuspension activity (Figure S10).

The deeper and thermally driven convective boundary layer during summer further increased turbulence, and vertical mixing helped to distribute the particles more evenly within the lower atmospheric layers, thereby homogenizing the particle concentrations. The agreement between summer campaigns with and without the dryer also confirms that under low-to-moderate RH conditions (< 50 %), humidity effects on eBC were minimal in the well-mixed daytime atmosphere. The lower eBC and PNC concentrations observed in summer 2024 compared with summer 2023 can be attributed to meteorology, measurement configuration, and emission changes. Firstly, during 2024, the wind speed (Figure S11) and boundary layer height (Figure S12) were higher, thereby enhancing ventilation and dilution of surface emissions. Secondly, a dryer was used with AE51 during the summer of 2024, thereby reducing the humidity-related overestimation that had affected the summer 2023 measurements. And finally, higher pollutant concentrations were measured in 2023 compared to 2024 in Prague, both by ground-level and 50 m PM data directly at the Prague site (Figure S13), and also at a nearby (2.3 km of the measurement site) CHMI national air quality network station Prague–Karlín station (Figure S14). This independent observation supports the UAV findings and confirms that the interannual difference primarily reflects meteorological variability and reduced local emissions in 2024.

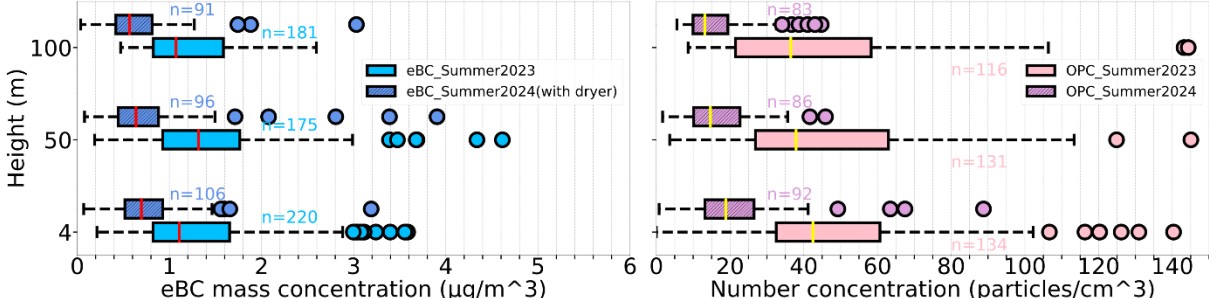

471

**Figure 7. Boxplots of (left) eBC concentration from AE51 without a dryer (summer 2023) vs. with a dryer (summer 2024) and (right) PNC from OPC at the urban site Prague during summer 2023 and 2024. Boxes show median and IQR; whiskers extend to 1.5×IQR; points beyond are outliers. n = number of points (1-min means) per altitude.**

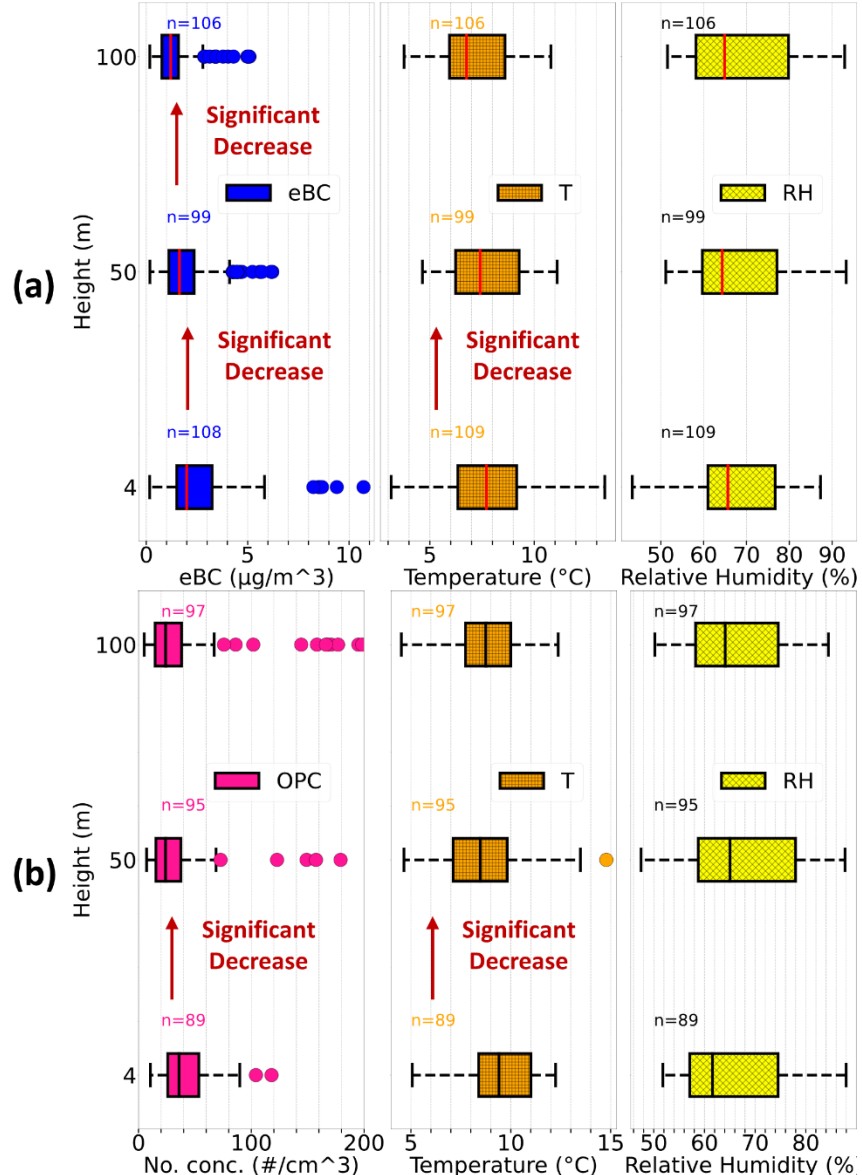

475

**Figure 8. Boxplots of (a) eBC mass concentration from AE51 without dryer vs. Temperature vs. RH, and (b) PNC from OPC vs. Temperature vs. RH on the drone at Prague from December 12 to December 18, 2023. Boxes show median and IQR; whiskers extend to 1.5×IQR; points beyond are outliers. n = number of points (1-min means) per altitude.**

During the winter campaign at Prague from December 12 to December 18, 2023, a significant reduction in both eBC mass concentration and PNC with increasing altitude was observed (Figure 8), contrasting with the summer pattern. Specifically, eBC mass concentration decreased with height up to 100 m, while PNC dropped to 50 m from the ground and remained constant between 50 and 100 m. This behavior is primarily influenced by the combination of strong emission sources in urban environments (Figure S10), as described previously, and the weak vertical temperature gradient ($\Delta T \approx 0.06$ °C between 4 m and 50 m), which does not support vertical mixing. During winter, reduced solar heating and long nocturnal cooling produce a shallow, near-isothermal boundary layer which suppresses turbulence and confines pollutants near the surface (Marucci and Carpentieri, 2019; Wang et al., 2018a). The resulting weak turbulent diffusion, rather than a distinct temperature inversion, explains the accumulation of eBC and PNC within the lowest tens of meters. The relatively smaller wind shear (1.1 m/s per 100 m between 50 – 100 m) further suppresses vertical mixing, trapping pollutants near the surface (Figure S11). As a result, pollution remains confined closer to the emission sources, leading to higher concentrations near the ground and a more pronounced decrease with height (Kotthaus et al., 2023). Additionally, the urban heat island effect intensifies during stable conditions, causing temperature contrast between urban and rural areas, further reducing the vertical dispersion of pollutants (Haeffelin et al., 2024).

Several outliers (extremely high levels) were detected in the eBC mass and PNC. During summer, the outliers can be linked to increased turbulences and daytime convective activities tend to flatten vertical gradients within the mixed layer, yet they can increase temporal variability at a fixed location by intermittently transporting near-surface plumes (e.g., traffic, cooking, construction) to the sampling height. Therefore, these outliers are episodic plume encounters rather than persistent stratification. Fewer outliers were observed at NAOK for eBC and PNC during winter, but more pronounced outliers were present in winter measurements at Prague. This high concentration was due to an elevated winter pollution event between December 13 and 14, 2023. This event was marked by a sharp rise in PM levels, as confirmed by low visibility signals from the drone at 100 m and ongoing PM measurements at the site (Figure S16). The vertical variation and other characteristics of this pollution episode were thus further studied to get a better understanding of the influence of such an event on air quality.

### 3.4. Vertical variation during an elevated winter pollution event in Prague

An increase in eBC mass concentration and PNC characterized Prague's winter pollution event in December 2023. The event started on December 13[th] at 13:00 and lasted until the morning of December 14[th], 2023. This concentration increase was primarily attributed to a low and stable boundary layer reaching 105 m above the ground (Figure S16). The vertical variation of eBC and PNC, along with the size distribution, was assessed to evaluate the changes one day before (i.e. December 12) and comprised 5 vertical profiles of eBC and 4 profiles of PNC. During the event, 3 profiles for both eBC and PNC were measured, all showing a substantial increase in concentrations at all heights (4m, 50m, and 100m) compared to the period before the pollution episode (Figure S17).

The highest increase in eBC concentrations during the event was observed at 100 m, with a 192% (2.5 $\mu g/m^3$) increase in median eBC levels compared to that before the event. Though less pronounced, the increase in eBC concentration was also seen at 50 m and 4 m, with 130% (1.5 $\mu g/m^3$) and 56% (1.7 $\mu g/m^3$) increase, respectively. The observed increase in eBC concentration at 100 m, just at the PBL height, suggests that while

ground-level emission had some impact, local atmospheric conditions allowed for some degree of vertical transport of eBC from the above layer, likely influenced by long-distance transported particles. This is supported by the back trajectory analyses, showing a change in trajectories from southwest to west at the beginning of the event, associated with transport of continental air masses from higher altitudes (Figure S18). The drone measurements not only support the measurements at the building (Figure S15), showing higher $PM_{2.5}$ and $PM_{10}$ concentrations at 50 m compared to 10 m results, but also provide measurements at 100 m, confirming the largest enhancement in eBC concentration during the event above the building compared to the ground and 50 m. Such vertical layering is consistent with many winter haze cases where polluted residual-layer air overlies a shallow, stagnant boundary layer, yielding dual source contributions—local near-surface emissions plus advected/aged aerosol aloft (Sun et al., 2016).

In contrast, PNC showed the highest increase at 4 m and 50 m in comparison to the day before the pollution event, where PNC increased by 840 % (238 particles/cm$^3$) and 860 % (151 #/cm$^3$), respectively, with a less pronounced increase at 100 m (460 %, 137 #/cm$^3$). This suggests that some particles, most likely generated from ground sources, remained concentrated near the surface due to the limited vertical dispersion during the pollution episode. Prior to the event, eBC concentrations exhibited a significant decrease (by 73 %) from the ground up to 100 m, and PNC decreased by 38 % between 4 m and 50 m. However, the trend was notably altered during the event, with no significant change in both eBC and PNC with the height, indicating that daytime mixing has weakened, allowing accumulation of pollutants throughout the shallow boundary layer.

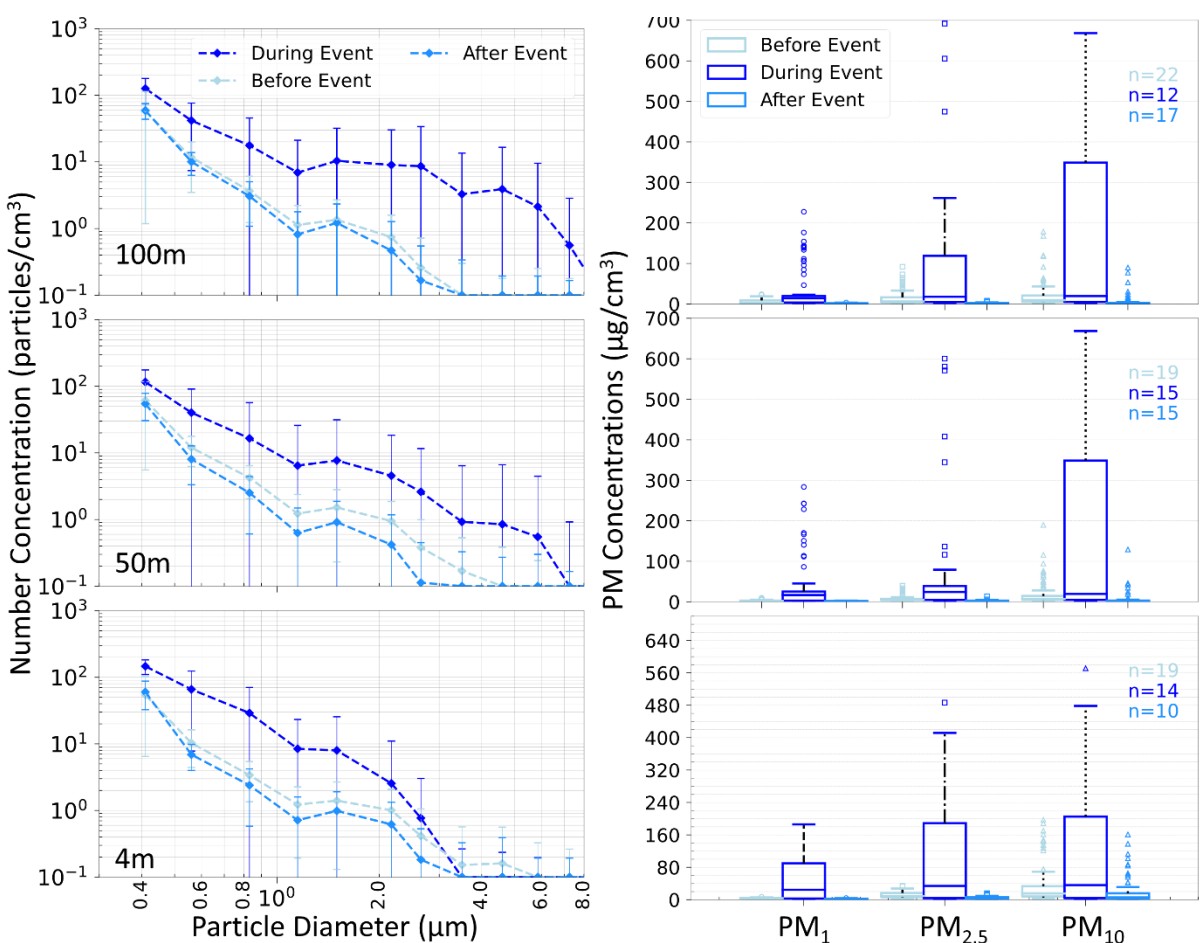

**Figure 9. Mean particle number concentration dependence on particle size from OPC on the drone at different heights before, during, and after an elevated winter 2023 pollution event in Prague. . n = number of points (1-min means) per altitude.**

The particle number size distributions at various heights reveal additional information (Figure 9). At 4m, the concentration of particles smaller than 3 μm increased significantly during the event compared to the distribution before the event, highlighting the production and accumulation of small particles near the ground. These particles likely stem from incomplete combustion and secondary formation under stagnant conditions, where condensation of semi-volatile vapors and coagulation processes enhance fine particle numbers (Gani et al., 2019; Zheng et al., 2023). In contrast, at 50 m and 100 m, concentrations of all particles were increased during the event, up to sizes of 10 μm. With the height, mainly the concentration of intermodal fraction, i.e., in sizes between 2.5 and 10 μm, increased in concentrations, potentially indicating contribution from longer distance transported aerosol and mixing processes that redistribute particles vertically. Before the event, particles up to 4 μm in diameter were observed near the ground due to winter stable atmospheric conditions (Gani et al., 2019), which restricted vertical mixing and limited dispersion of pollutants. During the event, a substantial increase in larger particles was observed at 100 m (and partly also at 50 m), while almost no change was observed at the ground level concentrations, remaining below 0.1 $\#/cm^3$, suggesting contributions from long-range transport disconnected from the ground. A significant increase in PM mass was also observed across all heights (4 m, 50 m, and 100 m) (Figure S19). $PM_1$ and $PM_{2.5}$ dominated the mass concentrations across all heights during the event, while $PM_{10}$ saw the largest increase at 100 m, again suggesting contributions from coarse particles and vertical mixing.

## 3.5.    Seasonal Contrast

The vertical profiles of eBC mass concentration and PNCs during the summer and winter campaigns were compared, revealing significant differences in the seasonal vertical patterns between the two stations.

At NAOK in winter, a 100% (0.45 μg/m³) higher eBC mass concentration up to 50 m was found compared to summer (Figure 10a). This difference can be attributed to more stable atmospheric conditions (isothermic to temperature inversion) hindering vertical mixing and to an increased number of sources during winter. At 100 m, the difference between winter and summer eBC mass concentration surged to 200% (0.5 μg/m³), as in summer, a decrease in eBC concentrations was observed above 50 m, while it was observed from 100 m in winter. The upward shift of the gradient during winter indicates a shallower mixed layer that traps pollutants within the lowest 100 m. Such seasonal layering of black-carbon aerosols has also been observed in at NAOK (Mbengue et al., 2020). In contrast, during the winter campaign at Prague, eBC mass concertation was 80% (0.88 μg/m³) higher at the ground level compared to summer, but the difference decreased to 24 % (0.32 μg/m³) at 50 m (Figure 10b). No significant difference in eBC mass concentrations in summer and winter was found at 100 m, indicating effective dispersion at this altitude at Prague, likely influenced by local factors such as the surrounding plateau, which alters airflow patterns and enhances the mixing of pollutants above the top of the valley. This suggests that while surface emissions dominate near ground level, mechanical turbulence generated by buildings and local topography enhances mixing aloft, mitigating vertical gradients

For PNC at NAOK, a 200% (19 #/cm³) increase was observed at ground level during winter compared to summer, which extended to 336 % (26 #/cm³) at 50 m and 200 % (148 #/cm³) at 100m (Figure 10c). However, at 150 m, the winter-summer difference was indistinguishable (differed only by 1 #/cm³) during the campaign, suggesting that particles disperse horizontally more rapidly than vertically at this altitude, likely due to atmospheric stability restricting vertical movements during winter. This is further supported by wind shear values between 100 – 150 m, which were 1.6 m/s per 100 m during summer, indicating higher turbulence and stronger vertical mixing. In contrast, winter exhibited lower wind shear (0.5 m/s per 100 m), signaling reduced turbulence and weaker vertical mixing (Figure S11), favoring horizontal dispersion over vertical transport.

In contrast, in Prague, the PNC behaved differently i.e., higher concentration was measured in summer compared to winter. The particle concentrations decreased with the height more during winter compared to summer (Figure 10d), with only a small difference at the ground level (15 %) (7 #/cm³) and a higher difference at 50 m and 100 m (36 %, i.e. 10 #/cm³ and 34 %, i.e. 12 #/cm³ respectively).

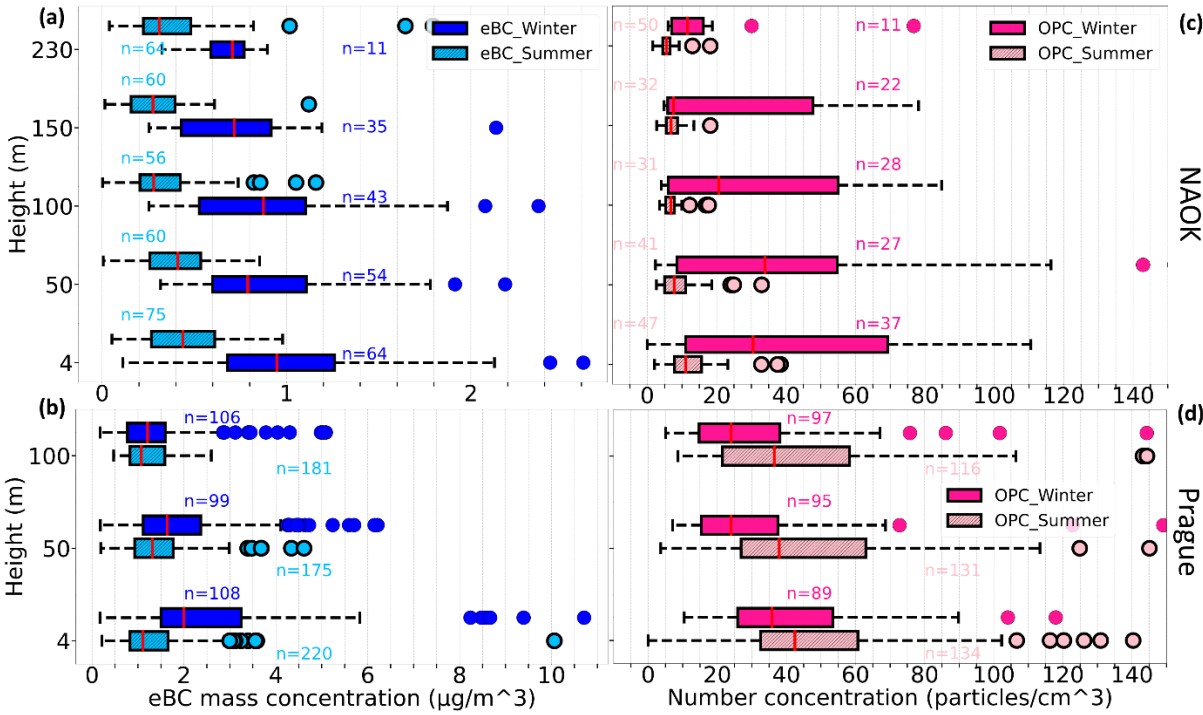

**Figure 10. Boxplots of eBC mass concentration from AE51 and PNC from OPC on the drone during summer 2023 and winter 2024 at (a, c) NAOK and summer 2023 and winter 2023 at (b, d) Prague. Boxes show median and IQR; whiskers extend to 1.5×IQR; points beyond are outliers. n = number of points (1-min means) per altitude.**

To understand these patterns further, particle size distribution was examined for the summer and winter campaigns at NAOK and Prague (Figure 11). It is important to note that the size distribution analysis excluded the high pollution event for Prague to avoid skewed results. At the NAOK site, both seasons showed a general decline in concentration as particle size increased; with winter concentrations consistently higher across all sizes (up to 3 μm), likely due to limited vertical mixing and increased combustion. As a result, in winter, the PNC over 1 #/cm³ were observed up to 1 μm at all heights, while in summer, the concentrations decreased below 1 #/cm³ for particles larger than 500 nm.

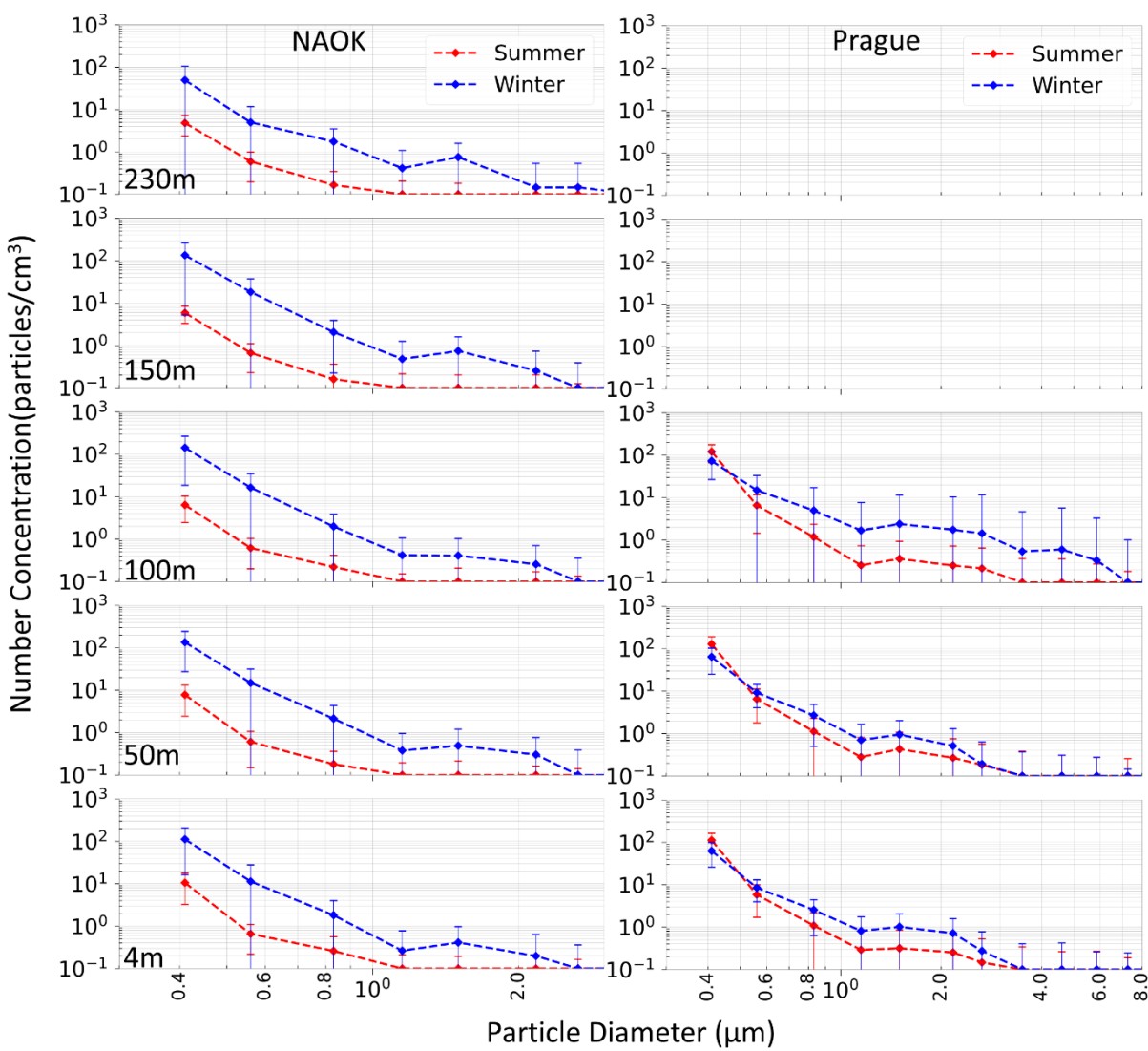

**Figure 11. Log-Log plot of the variation of mean particle number concentration with particle size from OPC on the drone at different heights during summer 2023 and winter 2024 at NAOK (left) and summer 2023 and winter 2023 at Prague (right).**

At the urban site, Prague, the size distribution analysis showed a significant increase in the average particle count for particles with sizes between 0.5 – 3 μm during the winter month (Dec, 2023) compared to the summer month (Aug, 2023) across all heights. Despite this increase, PNC was higher in summer than in winter (Figure 10d), due to higher summer concentrations in the smallest size bin, <0.5 μm (Figure 11). Although based on the limited number of flights and counts < 1 # cm$^{-3}$ for particles over 1 μm (Figure 11), this finding is consistent with ground-based observations from the national network (Figure S20), also showing higher concentrations of $PM_{10}$ in Aug 2023 than in Dec 2023.

OPC-N3 due to its detection limit of ~0.3 μm cannot directly capture new particle formation events and the subsequent growth that may be the reason for the increase in concentrations. The photochemistry-related origin of the summer aerosol in the smallest measurable bins is however supported by the higher concentration of nitrogen dioxide (NO2) during the summer (Figure S21), combined with increased sunlight (Figure S22), both of which promote the photochemical production of secondary particles (Gao et al., 2012; Kulmala et al., 2004).

The larger particles (2.5 - 8 μm) showed a more significant increase during winter in Prague, particularly
at 100m, further suggesting contributions from regional or long-range transported sources.
**4.      Summary and conclusions**
This study presents a campaign-based analysis of vertical measurements of eBC mass concentration and PNC
using drone-based profiling at a rural (NAOK) and an urban (Prague) site in the Czech Republic during different
seasons. A comparison of drone-deployed instruments with reference measurements at various heights of fixed
observational platforms (tall tower and building) was performed under various RH conditions and RH control
strategies.
The results show the effectiveness of drones for vertical profiling, offering results comparable to
reference instruments at various heights between 0 and 230 m and suggesting the applicability of drone eBC and
PNC measurements also in higher altitudes. When mounted on a drone, eBC mass concentrations from AE51 with
dryer were comparable at the ground and 230 m with the reference devices. Without the dryer, the eBC mass
concentration was overestimated by 276 % and 285% compared to the reference devices on the ground during
summer and winter, respectively, attributed to higher ambient RH levels. In comparison, results differ by less than
10 % from the reference when using a dryer. Thus, drying significantly reduces measurement discrepancies,
highlighting the importance of drying in minimizing the impact of RH, particularly for eBC measurements. While
the dryer study demonstrates close agreement on a single day, a multi-day validation with the dryer installed
remains a priority for future work, as our findings emphasize the necessity of a drying system even on drone-
based measurement platforms.
At the rural site (NAOK), eBC mass concentration and PNC decreased with height during both seasons,
though the height at which the decrease began was higher in winter than in summer. eBC mass concentrations
were uniformly distributed up to the first 50 m in summer and up to 100 m in winter. PNC decreased with height
from the ground in summer, while it remained uniform up to 50 m in winter, probably due to weak vertical
temperature gradients ($\Delta T \approx 0.05$ °C) and limited turbulent mixing during this season, which also led to higher
concentrations of both eBC and PNC compared to summer. The higher concentrations during winter at NAOK
were primarily driven by fine particles ($PM_1$) associated with combustion sources such as residential heating.
However, our results suggest that at least two aerosol populations of different sizes and sources were measured
during the year, thus with different vertical behaviors.
Conversely, at the urban site (Prague), both eBC and PNC were more uniform across altitudes in summer,
facilitated by local emission sources (supported by local air-quality data) and enhanced vertical mixing driven by
the urban heat island effect. eBC mass concentration and PNC in winter decrease with height, reflecting limited
vertical mixing due to near-isothermal conditions ($\Delta T \approx 0.06$ °C) and weak wind shear. PNC was higher in
summer, likely due to increased secondary particle formation driven by elevated levels of gaseous precursors and
photochemical reactions. These seasonal differences emphasize the interplay between emission strength,
boundary-layer dynamics, and secondary formation processes in shaping vertical aerosol patterns.
During a winter high pollution event in Prague, both eBC and PNC concentrations increased, with long-
range transport contributing to high eBC mass at 100m, while PNC remained concentrated near the surface. The

largest enhancement aloft coincided with the estimated boundary-layer top, suggesting entrainment of aged, transported aerosols above a shallow mixing layer, while near-surface PNC reflected trapped local emissions. These emphasize the dynamic interaction of local emissions, atmospheric stability, and long-range transport aerosols in shaping vertical concentration profiles, undecipherable by only ground-based measurements. Using drone-based measurements to capture vertical variation in air quality offers valuable insights into pollutant dynamics.

While the measurements presented here offer new insights into the vertical variability of eBC and PNC at rural and urban sites, they represent short-term case studies under specific meteorological and seasonal conditions. Therefore, the observed vertical structures and seasonal contrasts should be interpreted as site-specific patterns rather than generalized tendencies. Nonetheless, the study demonstrates the capability of UAV-based systems to capture vertical pollutant gradients with high spatial resolution, highlighting their potential for complementing long-term monitoring and model validation efforts.

**Author contribution**

KJ, DB, and NZ designed the experiments. KJ carried out all the experiments. KJ was also responsible for conceptualization, methodology, validation, formal analysis, investigation, data curation, visualization, and writing of the original draft. DB contributed to methodology and writing – review & editing. NZ was responsible for validation, supervision, and writing – review & editing. SM was responsible for data curation and contributed to writing – review & editing. VZ contributed to writing – review & editing, funding acquisition, and resources.

**Data availability**

The dataset including drone measured data and from reference devices and meteorological instruments, covering both rural and urban sites across different seasons is available at JULAHA, KAJAL (2025), "Drone_rural_urban", Mendeley Data, V1, doi: 10.17632/snbp6w49v9.1

**Acknowledgment**

We gratefully acknowledge the financial support provided by the National Research Infrastructure Support Project - ACTRIS Participation of the Czech Republic (ACTRIS-CZ LM2023030) and CzeCOS (LM2023048), funded by the Ministry of Education, Youth and Sports of the Czech Republic, and the Czech Science Foundation under grant 24-10768S, ICPF Internal Grant Agency (IGA - 028001) and Charles University Grant Agency (grant number 98124). We also thank colleagues from the Košetice observatory and Department of Atmospheric Physics, Prague, for providing the meteorological data. Special Thanks to our technicians, Petr Roztočil and Rohit, for preparing the Arduino datalogger and Airstream Dryer.

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
