# Peer review of "Drone-based vertical profiling of particulate matter size distribution and carbonaceous aerosols: urban vs. rural environment"

_EGUsphere, 2025_

## Referee Comment (RC1)

Work entitled "Drone-based vertical profiling of particulate matter size distribution and carbonaceous aerosols: urban vs. rural environment" is focused on important aspect of vertical profiling of atmospheric aerosols. It is still field, which lacks data, especially for absorbing aerosols, due to technical difficulties in obtaining it. Because of that every research contributing to available data sets is valuable and should be taken intro consideration. Presented paper confirms usefulness of remotely piloted aerial systems for aerosols studies. Interesting and worth sharing is concept of air dryer for a drone. This section could be very useful if will be properly supplemented. The work is trying to formulate general statements about nature of aerosols vertical variability in urban and rural environments during winter and summer, but despite unique dataset it could be hard to defend general thesis. Couple of sections and figures shows deficiencies in internal proof-reading and verification of presented study. In my option the work requires a major review. I hope my comments (attached) will help improve the paper and decrease doubt of potential readers. Good luck!

**114:** How reference of Petzold "Evaluation of Multiangle Absorption Photometry for Measuring Aerosol Light Absorption" 2005 is related to AE-51? More appropriate in context of eBC will be Petzold 2013 (doi:10.5194/acp-13-8365-2013), so maybe it is a mistake in reference.

**119:** AE51 reports absorbance ('Attenuation, ATN'), which is dimensionless, not in dB/m. I could recommend for aethalometer measurements and corrections details paper from Schmid et at. 2006 (www.atmos-chem-phys.net/6/3443/2006/). Please explain the value of ATN < 80. Add appropriate reference or explanation.

**2.2.1. eBC measurements**: General question to the processing of AE51 data. This device is known for reporting negative eBC values in pristine conditions. Data collected during presented study has occasionally negative values of eBC too. How this issue was addressed? In different studies different approaches were used to deal with that issue, usually they are based on assumption of monotonically increase of ATN over time.

**2.2.2. Air stream Dryer:** Using build in-house air stream dryer on UAV is valuable and interesting approach. It is worth describing in details results obtained during laboratory tests.

**124-125:** What was size of the mesh grid? How minimal particle loss was evaluated?

**126-128:** What was the performance of the drier with flow rates characteristics for used sensors (how changed humidity from ambient to conditions inside an instrument)? Flow tests are mentioned, but their results are not presented. With small sensors problem of low flow is well known, what is confirmed in section dedicated to OPC-N3, so it is interesting to know this characteristic of the presented dryer.

**131:** PSL acronym (planetary surface layer) not explained before

**134-136:** Are temperature sensors mounted in the AQ backpack on the same side or on different sides of it? In case of different sides, is there any solution for compensation/covering from heating of sensors because of sun location (heating of sensor / sensor housing / drone hull)?

**138:** More than sensors accuracy evaluation it was evaluation of feasibility of temperature vertical profiles measurements with the presented set-up.

**139 – 142**: Existence of correlation between temperature measurements on UAV and the tower is quite obvious, but definitely missing is information on errors and bias of measurements. In supplement are 30 panels with measurement results with fits and R, but they should be summarised.

Are there any tendencies? Thermometers on UAV are showing higher, lower, similar temperatures in comparison to stationary measurement?

Please comment on very low correlation values mentioned in the paragraph. Why measurements of the same physical quantity: temperature / relative humidity are so different on UAV and the tower? As the paper aims to presents how valuable are drone-based measurement it is worth to discuss more on their limitations and potential sources of uncertainties.

**142 – 144**: On presented Fig. 2 is visible that both AE51 and OPC-N3 have horizontal inlets, so why only in OPC-N3 it is impossible to use dryer?

**143:** OPC-N3 flow is driven by a ventilator, which was exposed directly to ambient conditions. How horizontal wind influenced results depending on wind direction in relation to UAV heading (ventilator front/side/back to the direction of wind)?

**149:** Alphasense in OPC-N3 specification shows sample flow of 0.280 l/min. Is the version used in this study different (0.21 ml/min)?

**153:** Why different temperature/rh sensors were used for OPC and AE51? What was a reason for using HYT939p, which has response time usually unsuitable for UAV operation (slow sensor with response time of couple of seconds)?

**161:** Figures in supplement lack reference to sensor model in caption. Once again it would be nice to have more details about comparison results with tower measurements, not only correlation.

**Figure 2.** In the pictures (f) and (e) it looks that inlet geometry was changed between version with and without dryer. In version with dryer we see that there are probably two $90^{\circ}$ bends of the inlet, while without dryer I see only one. Was placement of AE51 changed between designs?

**173 – 176:** Which version of Particle Loss Calculator by Weiden et al. was used for estimation of inlet losses? Could you provide geometry of inlets in supplement to reproduce calculations?

**187 – 192:** In which section continuous measurements in the whole profiles are presented? I see only stationary results from mentioned different heights. Could you more elaborate on mentioned propellers effects on aerosols flow during descend flights (reference)?

**190:** Should I understand that results presented in chapter 3 are measured during hovering (3 – 5 minutes) and then averaged for the whole period of hovering or just all data points were added to one large dataset grouped by altitude and then medians are only presented? This is quite important information regarding data processing.

**197-199:** Is it local time or UTC time? I suggest using 24 h time format, as it is easier to look into data files, where most of the results are reported in UTC time 24 h format.

**3. Results and Discussion:** Presented results definitely lack estimation of uncertainties, both for relative and absolute differences. Uncertainty analysis should be done in regards to aerosol conditions. In pristine air masses with low concentration of aerosols, accuracy of measurement technique already adds significant part of the final result.

As I already mentioned at the line 190 without knowing the data processing scheme is it hard to fully understand presented data. Are the presented boxplots made from all data points collected across all flights during the whole season or each flight it's statistics is a single point.

In general in the whole discussion I see no reference to any supporting observations made in both sites. All statements on general nature, which are made in the paper should be supported. Ground validation from established devices will be very useful. I have a feeling that in the paper are mixed two goals (a) to show that UAV-based measurements are well aligned with observation made from fixed platforms (towers) and (b) to observe general phenomenons related to vertical variability of aerosols. In a result both goals are missing strong support from evidence.

**243:** On 31/07 – 01/08 humidity was below 50%, as on 04/08. Is it reflected in data, that the differences were smaller and comparable between heights?

**254:** Why the first size bin of OPC was skipped? Most of presented results are showing very low PNC, so skipping the first bin, which usually has to most of them removes most of variability, which could be observed. Assuming OPS was the reference devices, based on size distribution from it was possible to have comparable size ranges for both devices, without removing data from the dataset.

**257:** Theoretically OPC reports airflow speed. Were you able to verify how stable was the airflow with different orientation of the drone against the wind?

**Figure 3:** It would be easier to read the figure if grid (vertical, axis X) will be visible. How outlier were determined? In 3a and 3c we see that outliers of AE33 are close to the median of AE51. Are those outliers single data points or data for a whole flights (it means which fraction of results are they representing)? In caption year of measurements is missing.

**277-279:** How a single day of flights (with probably quite similar aerosol conditions) confirms effectiveness in compression with long-term tower observations. I think it is an overestimation. It proves that during that particular day it was possible to obtain comparable results.

**291:** Why not parametric Kruskal-Wallis test was selected? Does it mean distribution of eBC and PNC results was expected to be not normal? What are the foundations for that assumption?

**Figure 5:** In 5c, similary as in Fig. 3 outliers from AE33 are surprisingly close to each other and close to results from AE51. Year missing in caption.

**298-300:** Uniform temperature usually means that layer is just very well mixed and if we have mixing, then it is not a stable layer. In this type of research stable conditions are usually connected with occurrence of temperature inversion, what is mentioned in cited paper by (Altstädter et al., 2020).

**305:** It is true that we could expect eBC mostly in the first bin of OPC-N3, which was excluded from the analysis, but as flights were done in a different time it could just mean, that different air mas was measured.

**315:** Is the statement about different aerosols populations confirmed by reference devices installed on the tower? That should be a point of reference.

**325:** If the wind sheer was 20 m/s per 100 m was it still possible to operate the UAV? Mavic 3 classic wind resistance given by DJI is 12 m/s, but here change was much above that. From the figure S9 I am not able to reproduce value 20 m/s.

**Figure 7:** Presented temperatures on both panels suggest that in December 2023 in Prague was 28$^{\text{O}}$C. Does not look realistic, probably messed up with data from summer.

**339 – 340:** As mentioned before limited mixing and stability of layers, leading to accumulation of aerosols could be easily observed by temperature inversion, which is not visible in the presented data.

**346-348:** Increased turbulence and convective activities are both factors known for strengthening of mixing, especially when we assume local, near to the ground sources of aerosols. Cited paper "Assessing the Internal Variability of Large-Eddy Simulations for Microscale Pollutant Dispersion Prediction in an Idealized Urban Environment" is dedicated to assessment of LES models and do not confirms that turbulence and convection increase vertical variability of aerosols concentration.

**355:** It could be worth adding some reference point to call the event highly polluted – monthly mean, or some other reference. Presented values suggest that for European conditions it was not so polluted, as there are papers showing measured values over 50 µg/m$^3$.

**365 - 380:** Low PBL height usually means that there is higher accumulation of aerosols closer to the ground, what is quite typical for higher concentration during winter, when their source is mostly in combustion/emission of fine mode of particles (like eBC). Presented results suggest that source of aerosols is different, with mostly course mode particles. It is interesting observation and I would be valuable to estimate potential source of observed aerosols. Fast transport mentioned with back-trajectories is indeed more favourable for different sources of aerosols than those typical for winter, related to anthropogenic emissions. Could you try to tell more about this air mas? What kind of aerosols it was and from where originating? Was it mineral dust or sea salt? AERONET, CALYPSO and modells data could be helpful here.

**399 – 400:** I would be careful about general statement on seasonal vertical patterns, when the reasoning is based only on such limited dataset.
**413:** Presented difference of just one particle in cubic centimetre is probably still in uncertainty of used instrument. In general all comparisons when PNC is 10 is quite hard, knowing how problematic is signal-to-noise ratio in micro optical particle counters.

**432 – 434:** It is not consistent if we have in winter mostly fine mode particles and then as a contrast is presented summer with "smaller particle sizes". Please verify that sentence.

**441 – 442:** As mentioned by authors observation of new particles formation with OPC-N3 is quite unrealistic. Higher concentrations in summer is an interesting observation, but once again it is important to remember that we are working on a limited dataset with relatively low concentration at all. Figure 11 shows that most of the bins are below 1 #/cm$^3$. I would be cautious with presented statement and look for additional ground reference for seasonal variability.

**Summary:** The main problem, which I see is that we have mixed statements trying to be general with, those more related to a case study observations. Presented data is valuable due to general lack of vertical profiling, what was mentioned in the introduction, but it's values could not be overestimated in suggesting general tendencies in places, where data was collected.

**449 – 450**: I would not call analysis made during 27 days split into two sites and two seasons "comprehensive".

**473:** Which sources are called here "strong"? Do you have access to results from ground base air quality network, which could support statements on local emissions in the city?